# SHAP-Guided Feature Selection and Model Simplification for Facial Time-Series Reaction Detection

**Andrei Kulinskii**
Russian Academy of Science (RAS)
Moscow, Russia
layneys13@gmail.com

**Ilia Stulov**
Russian Academy of Science (RAS)
Moscow, Russia
ilya.stulov.2501@gmail.com

**Bratenkov Miron Andreevich**
Novosibirsk State University (NSU)
Novosibirsk, Russia
m.bratenkov@g.nsu.ru

**Olga Krivorotko, Konstantin Gnidko**
Sirius University
Sirius Federal Territory, Russia
{krivorotko.oi, gnidko.ko}@talantiuspeh.ru

**Sergei Strijhak**
Russian Academy of Science (RAS)
Moscow, Russia
strijhak@yandex.ru

## Abstract

Facial micro-reactions provide a non-invasive signal for detecting short-lived emotional responses during constrained question–answer episodes. However, raw FaceMesh landmark trajectories are high-dimensional and strongly correlated, which complicates learning and interpretation. We build an interpretable facial time-series baseline by mapping FaceMesh dynamics to a compact set of 17 semantically meaningful features (e.g., eye and mouth openness, symmetry, and motion descriptors) and train a Transformer-based classifier to predict a binary target (reaction vs no reaction).

To explain model decisions and quantify feature contributions, we use SHAP. In practice, we compute attributions via Deep SHAP, which combines DeepLIFT-style reference-based propagation with Shapley-value principles to obtain efficient, structured approximations for deep networks. Beyond post-hoc explanation, we introduce an iterative SHAP-guided feature selection procedure: we train the model, estimate global importance via mean absolute SHAP values, prune low-impact or unstable features, retrain, and reassess predictive ability. This yields a more compact representation aimed at preserving accuracy while reducing input dimensionality.

## 1 Introduction

Inferring a person's internal state from outward behavior is a notoriously difficult problem: even in highly structured settings, subtle and short-lived cues are easy to miss, inconsistent across individuals, and sensitive to context. A practical and comparatively non-invasive alternative to instrumented physiological sensing is video-based facial analysis, which can be collected with a single camera during constrained "question–answer" episodes such as interviews, screenings, or structured assessments (Islam et al., 2021), (Nam et al., 2023).

Recent work highlights the importance of multimodal analysis for understanding human behavior and deception. For instance, a large-scale study on the Indian population introduced a multimodal dataset combining seven modalities, including EEG, ECG, EOG, eye-gaze, GSR, audio, and video, and demonstrated that joint modeling of behavioral and physiological signals can significantly improve detection performance compared to unimodal approaches (Joshi et al., 2025). Similarly, a

multimodal framework developed on a Colombian dataset integrates visual, audio, and language-based reasoning components (e.g., ViViT, HuBERT, and LLM), showing that combining perceptual signals with high-level semantic reasoning leads to more robust and interpretable deception detection systems (Grabowski et al., 2025).

In this paper we focus on reaction detection (presence/absence of a measurable facial response to a prompt), rather than attempting to infer veracity; importantly, a detected reaction should be interpreted as salience or emotional involvement, not as evidence of deception. Facial behavior is an information-dense channel of human communication. Research in psychology distinguishes more sustained macro-expressions from micro-expressions, which can occur within fractions of a second and are often difficult to perceive reliably without assistance (Yildirim et al., 2023), (Monaro et al., 2022). A widely used descriptive framework is the Facial Action Coding System (FACS) (Ekman & Friesen, 1978), which represents facial activity through anatomically grounded Action Units corresponding to muscle activations. While classic approaches relied on manual coding, modern computer vision pipelines enable large-scale, automated analysis of facial motion, typically by detecting the face region and extracting a dense set of facial landmarks. One prominent example is MediaPipe Face Mesh, which estimates 468 facial landmarks in real time from a single RGB camera input.

Despite the availability of dense landmark trajectories, directly training deep models on raw landmark coordinates is often inefficient and hard to interpret. The input is high-dimensional, highly correlated, and sensitive to nuisance factors such as camera position, head pose, and illumination, which can inflate model capacity requirements and complicate reproducibility. Instead, we adopt a structured representation that maps Face Mesh trajectories into a compact set of 17 semantically meaningful facial dynamics features (e.g., eye openness and blink-related dynamics, mouth-related dynamics, symmetry and motion-derived descriptors). These features are designed to preserve the temporal structure of facial behavior while reducing redundancy and improving interpretability.

The Transformer architecture was originally introduced in "Attention Is All You Need (Vaswani et al., 2023)", where sequence modeling is performed via self-attention rather than recurrent or convolutional operations. This work implements a Transformer Encoder classifier using PyTorch and configure it specifically for facial dynamics time-series. However, even with a compact representation, it remains essential to determine which features are truly driving model decisions, and whether some features are redundant or noisy. For this purpose we use SHAP (SHapley Additive exPlanations), which provides feature attributions grounded in cooperative game theory through Shapley values.

Crucially, we use SHAP not only for post-hoc explanation, but as a model simplification tool. We propose an iterative SHAP-guided feature selection loop: train a baseline transformer on all 17 features, compute global feature importance (e.g., mean absolute SHAP values), remove low-contribution and unstable features, retrain, and re-evaluate predictive ability and stability. In the planned staged procedure, we first reduce the feature set from 17 to 12, recompute SHAP on the pruned model, and then explore a further reduction to 11 features, repeating the same cycle to assess convergence of feature rankings and prediction ability. The resulting compact model is expected to improve robustness by discarding noise-dominated inputs, and provide clearer scientific insight into which facial dynamics are most associated with reaction events. The full pipeline is presented on the next figure 1:

## 2 DATASET

The raw data consist of per-subject session recordings captured in the MTS format, i.e., an MPEG-2 Transport Stream (MPEG-TS) container, with the following streams:

- Video: H.264 (High Profile), 1920×1080, 25 fps;
- Audio: AC-3, 48 kHz, 5.1 channels;
- Additional: PGS subtitle stream (hdmv_pgs_subtitle);
- A typical input file has a duration of about 5 minutes;
- A file size ranging from 300 to 700 MB

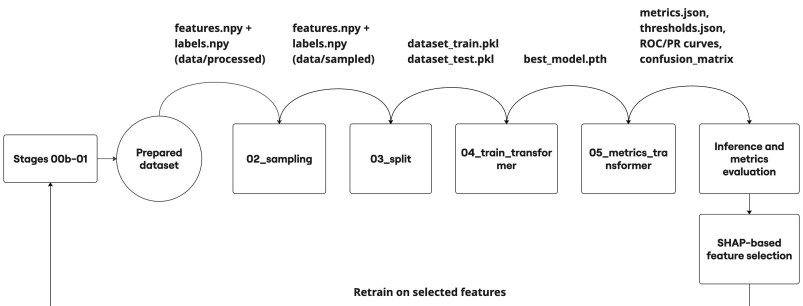

Figure 1: The whole pipeline.

For each participant, one recording is typically 4–5 minutes long; interrupted or incomplete sessions may be shorter.

Although MTS is common for professional video capture, it is not supported by most web players and browser-based annotation tools, including the Label Studio interface. In addition, the embedded AC-3 audio codec is not natively supported by standard browsers. As a result, manual labeling in the original format was impractical.

To ensure compatibility with Label Studio, the raw MTS files were converted to MP4 using FFmpeg with the following settings:

- Video: H.264 (High Profile), 1920×1080, 25 fps;
- Audio: AAC-LC, 48 kHz, stereo.

The AAC-LC codec was chosen due to its cross-browser support in HTML5 players. This conversion enabled (i) correct audio playback during annotation, (ii) stable audio–video synchronization, and (iii) precise identification of question onsets and the subject's response timing. The processed MP4 files typically have a duration of 3–5 minutes and a size of 300–700 MB. Importantly, audio quality is not critical for the current facial-model training, but it is critical for manual annotation because annotators must clearly hear the question to align the labels with the protocol.

In addition to the video material, each subject/session is accompanied by:

1. A DEX file containing raw physiological signals recorded by the Face Russian Truth Machine (FRTM) during the session (e.g., respiration, pulse, blood pressure, etc.). These signals are not used in the current training set but are reserved as a potential source for future multimodal models.

2. A printed protocol report in pdf/docx format, summarizing the session and indicating questions where a reaction was noted. The protocol is used during labeling to verify segments corresponding to the positive (reaction) label. One protocol is provided per subject.

From the perspective of raw data organization, one subject typically has:

- 1–2 primary session recordings (MTS);
- 1–4 auxiliary recordings;
- 1 DEX file (physiological data);
- 1 printed protocol report.

After conversion and removal of noisy or irrelevant material, the training set includes only pre-test and/or main-test recordings in which the subject sits steadily and looks directly at the camera. Video recordings were collected under controlled conditions. Participants seated directly in front of the camera with their faces illuminated by room lighting. To ensure reliable face tracking and consistent landmark extraction, subjects (participants) were instructed to face the camera and avoid significant head rotation or sideways positioning.

Pairs of question-answer segments were labeled using the Label Studio annotation tool. The annotation scheme included three labels: question, truth, and lie. These labels were sufficient for constructing the training dataset used in the current study.

But it is important to know that as a result of prediction, the definition of the target variable was intentionally formulated as "reaction/no reaction" rather than "truth/lie." A reaction does not necessarily indicate deception, it should be highlighted. Instead, it may reflect emotional stress, discomfort with a topic, partial disclosure, or other psychological factors. In practice, the presence of a reaction indicates that the interviewer should ask additional clarifying questions to better understand the situation.

Taking into account the distribution of the labeled events, the dataset contains 495 pairs of question-truth and 45 pairs of question-lie. These results are in a strong class imbalance. To mitigate this issue during model development, sampling procedures were applied in the training pipeline.

For evaluation purposes, 20% of the dataset were reserved as a test set. Due to the strong class imbalance, sampling was first applied to construct a balanced training subset, after which the final separation into training and test sets was performed.

Overall, the data were collected in 2023 for 44 participants (May–July) and 34 participants (February–April). For the May–July period, 77 videos were collected; only 29 videos from 22 participants were deemed informative enough for model training. For the February–April period, 95 videos were processed; 31 videos (approximately 25 participants) were retained as useful. In total, the final training dataset includes 47 participants and 60 videos.

## 3 IMPLEMENTATION AND PROCESSING PIPELINE

This section briefly describes the end-to-end software pipeline used to transform raw interview recordings and manual annotations into a reproducible training and evaluation workflow for facial time-series reaction detection. The pipeline is implemented in Python 3.10.x and organized as a DVC-managed DAG, with Hydra used as a unified configuration layer for stage parameters (paths, clip durations, sampling ratios, model hyperparameters, etc.). Video I/O and preprocessing are handled via OpenCV and FFmpeg; numerical processing relies on NumPy/SciPy and pandas; model training and inference are implemented in PyTorch; evaluation uses scikit-learn; and visualizations are produced with matplotlib and seaborn. Intermediate artifacts are serialized using JSON, pickle, and NumPy npy/npz formats.

### 3.1 MODELS

We use a Transformer-based classifier to process temporal sequences of facial-behavior features extracted from video; the main hyperparameters are reported in Table 1. For each frame, YOLO and MediaPipe FaceMesh produce a numeric feature vector (e.g., landmark geometry and simple derived dynamics). The resulting per-frame vectors form a fixed-length sequence that is first projected from the input feature space to a latent representation and then combined with sinusoidal positional encoding. A learnable CLS token is prepended to the sequence, and the resulting token stream is processed by a multi-head Transformer encoder.

The final prediction is produced from the encoded CLS representation using an MLP classification head (two linear layers with GELU and dropout), which outputs a single logit for binary classification; the head hyperparameters are summarized in Table 2. In this model, $d_{model}$ denotes the dimensionality of the internal token representation used throughout the encoder (i.e., the hidden embedding size): inputs are projected to $d_{model}$, and all attention and feed-forward sublayers operate on vectors of this size.

### 3.2 FACE FEATURES

Face localization is performed using YOLO (Ultralytics v8) as the primary detector due to its mature pre-/post-processing utilities and the option to fine-tune on domain-specific data if required. Landmark extraction is performed with MediaPipe Face Mesh, which produces dense facial landmarks per frame; the YOLO detector is used to localize and crop the face before Face Mesh inference (al-

Table 1: Transformer encoder hyperparameters

| HYPERPARAMETER | VALUE |
|---|---|
| $d_{model}$ | 128 |
| Number of encoder layers | 2 |
| Number of attention heads | 4 |
| FFN hidden size | 256 |
| Dropout | 0.1 |
| Activation | GELU |

Table 2: MLP head hyperparameters

| HYPERPARAMETER | VALUE |
|---|---|
| Input dimensionality ($d_{model}$) | 128 |
| Number of hidden layers | 1 |
| Hidden layer size | 128 |
| Activation | GELU |
| Dropout | 0.1 |
| Output | 1 logit |

ternatively, MediaPipe Face Detection can be used as a detector). The full landmark mask is shown on the figure 2 that is taken from the site: `https://ai.google.dev/edge/mediapipe/solutions/vision/face_landmarker`.

For each frame, the face is represented by a set of approximately 450–480 landmarks, corresponding to roughly 900 two-dimensional coordinate features $(x, y)$ per frame. This representation defines the core feature space for subsequent processing steps, including:

- aggregation of landmark points by anatomical regions (eyes, lips, eyebrows, etc.);
- computation of derived features (angles, distances, relative displacements);
- temporal averaging and normalization within a question–answer clip.

Table 3 lists the landmark groups (first and second columns) and their corresponding MediaPipe Face Mesh indices (third column) used during tracking and feature construction for model training.

### 3.3 KEY PIPELINE STAGES

1. Annotation conversion to QA JSON (stages/00b_process_annotations).
   Manual labeling is converted into a unified "question-answer" JSON format that standardizes the time intervals for questions, answers, and paired segments. This representation is used downstream for fully automated clip extraction.

2. Clip extraction (stages/00c_extract_clips).
   Using the QA JSON, the system automatically cuts video into short clips in three modes: question-only, answer-only, and paired question+answer segments. This step standardizes training examples and increases the number of samples per raw recording.

3. Face detection and landmark extraction (stages/00d_face_mesh).
   For each frame in each clip, the face region is detected, cropped, and normalized to a fixed resolution (512×512). The primary face detector is YOLO (Ultralytics v8) (optionally MediaPipe Face Detection). After face normalization, MediaPipe Face Mesh is applied to extract per-frame facial landmarks. The output of this stage is a frame-aligned landmark sequence for each clip.

4. Per-frame feature computation (stages/00e_features).
   Raw landmarks are transformed into 17 stable, interpretable, and scale-invariant facial dynamics features. These features are designed to be robust to nuisance variability and to

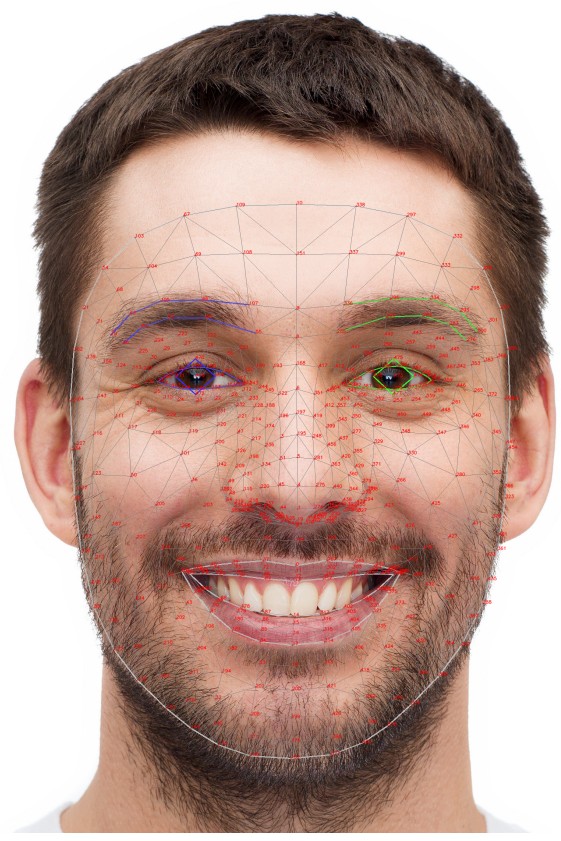

Figure 2: A landmark mask applied to a person's face as a result of Face Mesh stage.

Table 3: Landmark group table (MediaPipe Face Mesh indices) used for feature construction.

| GROUP NO. | GROUP/ZONE | MEDIAPIPE FACEMESH INDICES |
|---|---|---|
| 1 | Left eye (eyelid contour) | 33, 160, 158, 133, 153, 144 |
| 2 | Right eye (eyelid contour) | 263, 387, 385, 362, 380, 373 |
| 3 | Left eye center | 33, 133 |
| 4 | Right eye center | 263, 362 |
| 5 | Mouth corners | 61, 291 |
| 6 | Mouth vertical points (upper/lower) | 13, 14 |
| 7 | Left cheek (facial boundary) | 234 |
| 8 | Right cheek (facial boundary) | 454 |
| 9 | Nose tip | 1 |
| 10 | Left eyebrow | 70, 105, 66 |
| 11 | Right eyebrow | 336, 334, 296 |
| 12 | Left iris/pupil | 468, 469, 470, 471, 472 |
| 13 | Right iris/pupil | 473, 474, 475, 476, 477 |

preserve the temporal structure relevant for reaction detection (e.g., eye and mouth dynamics, symmetry and motion descriptors).

5. Tensor dataset construction (stages/01_build_dataset).
   Per-frame CSV feature files are converted into compact tensor representations and auxiliary structures required for training. Sequences are padded to a common length L (equal to the maximum sequence length within the dataset or batch, depending on configuration) and accompanied by a mask.

6. Class sampling (stages/02_sampling).
   A critical factor in data preparation is class imbalance. In contrast to another work De Marsico M. (2024), where classes are nearly balanced (a class "lie" has 162 samples and a class "truth" has 163 samples), our data exhibit a strong skew: class 0 ("no reaction") contains 495 samples, while class 1 ("reaction") contains 45 samples. To mitigate this, we implement an under-sampling strategy that reduces the majority class until the share of class 1 matches a target ratio defined in the configuration. In practice, balancing leads to more stable gradients during training.

7. Train/validation split (stages/03_split).
   To ensure valid learning and unbiased evaluation, we perform a stratified split into independent training and validation sets. The split is fixed prior to training so that the validation set does not influence optimization, while stratification preserves the minority-class ratio across subsets.

8. Model training (stages/04_train_transformer).
   A lightweight Transformer Encoder classifier is trained for binary reaction prediction. The model takes a sequence of per-frame features as input and outputs the probability of a reaction event. Training follows a standard binary classification setup with binary cross-entropy as the loss function.

9. Metric computation and threshold optimization (stages/05_metrics_transformer).
   We compute both threshold-dependent metrics (precision, recall, F1, balanced accuracy, specificity, accuracy, MCC, NPV, TPR, FPR) and threshold-independent metrics (ROC-AUC, PR-AUC, Brier score) for the best checkpoint. The decision threshold is selected using Youden's index.

## 4 SHAP: SHAPLEY ADDITIVE EXPLANATIONS

### 4.1 ADDITIVE FEATURE ATTRIBUTION METHODS

Complex models, especially in deep learning, are hard to interpret directly (Lundberg & Lee, 2017). The solution is to construct an *explanation model* $g$ that approximates the original model $f$ locally. Let $z' \in \{0, 1\}^M$ be a simplified binary input mapped to the original space via $x = h_x(x')$. Additive feature attribution methods define $g$ as a linear function of these binary variables:

$$g(z') = \phi_0 + \sum_{i=1}^{M} \phi_i z_i', \tag{1}$$

where $\phi_i \in \mathbb{R}$ is the attribution of feature $i$, and summing all attributions approximates the output $f(x)$.

### 4.2 DEEPLIFT

*DeepLIFT* is a recursive explanation method for deep learning that assigns each input $x_i$ a contribution score $C_{\Delta x_i \Delta o}$ measuring the effect of shifting $x_i$ from a reference value $r_i$ to its actual value. It enforces a *summation-to-delta* property:

$$\sum_{i=1}^{n} C_{\Delta x_i \Delta o} = \Delta o, \tag{2}$$

where $\Delta o = f(x) - f(r)$. Setting $\phi_i = C_{\Delta x_i \Delta o}$ and $\phi_0 = f(r)$ shows that DeepLIFT satisfies Equation 1, making it an instance of an additive feature attribution method.

### 4.3 CLASSIC SHAPLEY VALUE ESTIMATION

Three methods ground feature attribution in cooperative game theory (Li et al., 2024).

*Shapley regression values* quantify each feature $i$'s importance as its average marginal contribution across all subsets $S \subseteq F \setminus \{i\}$:

$$\phi_i = \sum_{S \subseteq F \setminus \{i\}} \frac{|S|! \, (|F| - |S| - 1)!}{|F|!} \left[ f_{S \cup \{i\}}(x_{S \cup \{i\}}) - f_S(x_S) \right]. \tag{3}$$

Setting $\phi_0 = f_\emptyset(\emptyset)$ makes this an additive feature attribution method satisfying Equation 1.

*Shapley sampling values* extend Equation 3 to black-box models by (1) sampling-based estimation and (2) approximating feature removal by integrating over the training distribution, avoiding the exponential $2^{|F|}$ cost of exact computation.

*Quantitative Input Influence* is a broader interpretability framework that independently derives a nearly equivalent sampling-based Shapley approximation, thus also qualifying as an additive feature attribution method.

### 4.4 PROPERTIES FOR UNIQUE DETERMINATION OF ADDITIVE FEATURE ATTRIBUTIONS

There exists a unique solution to additive feature attribution satisfying three properties.

*Local accuracy* requires the explanation model to match $f(x)$:

$$f(x) = g(x') = \phi_0 + \sum_{i=1}^{M} \phi_i x_i'. \tag{4}$$

*Missingness* requires absent features to carry zero attribution:

$$x_i' = 0 \implies \phi_i = 0. \tag{5}$$

*Consistency* requires that if a feature's marginal contribution increases under a model change, its attribution must not decrease: if $f_x'(z') - f_x'(z' \setminus i) \geq f_x(z') - f_x(z' \setminus i)$ for all $z'$, then $\phi_i(f', x) \geq \phi_i(f, x)$.

The unique solution satisfying all three properties is:

$$\phi_i(f, x) = \sum_{z' \subseteq x'} \frac{|z'|! \, (M - |z'| - 1)!}{M!} \left[ f_x(z') - f_x(z' \setminus i) \right]. \tag{6}$$

Any additive attribution method not grounded in Shapley values necessarily violates local accuracy and/or consistency.

### 4.5 SHAP (SHAPLEY ADDITIVE EXPLANATION) VALUES

SHAP values are the Shapley values of the conditional expectation $f_x(z') = E[f(z) \mid z_S]$, where $S$ is the set of non-zero indices in $z'$. Since most models cannot handle missing inputs, this expectation is approximated via a chain of simplifying assumptions:

$$f(h_x(z')) = E[f(z) \mid z_S] \qquad \text{(SHAP mapping)} \tag{7}$$
$$\approx E_{z_{\bar{S}}}[f(z)] \qquad \text{(feature independence)} \tag{8}$$
$$\approx f([z_S, E[z_{\bar{S}}]]). \qquad \text{(model linearity)} \tag{9}$$

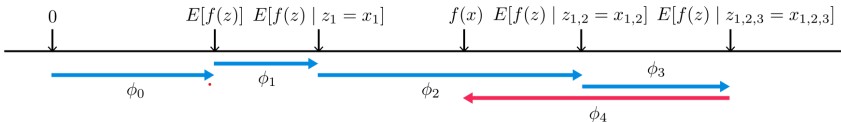

Figure 3: SHAP values attribute to each feature the change in expected model prediction when conditioning on that feature, showing how to transition from the base value $E[f(z)]$ to the current output $f(x)$. For non-linear models or dependent features, SHAP values are averaged across all possible feature orderings.

### 4.6 DEEP SHAP (DEEPLIFT + SHAPLEY VALUES)

Interpreting DeepLIFT's reference value as $E[x]$ reveals that it approximates SHAP values under feature independence and model linearity via linear composition of non-linear components. Since

DeepLIFT satisfies local accuracy and missingness but not necessarily consistency, it is natural to adapt it into a compositional SHAP approximation — *Deep SHAP*.

Deep SHAP propagates DeepLIFT multipliers expressed in terms of SHAP values backward through the network via a chain rule:

$$m_{x_j f_3} = \frac{\phi_i(f_3, x)}{x_j - E[x_j]}, \qquad m_{y_i f_j} = \frac{\phi_i(f_j, y)}{y_i - E[y_i]}, \tag{10}$$

$$m_{y_i f_3} = \sum_{j=1}^{2} m_{y_i f_j} m_{x_j f_3}, \qquad \phi_i(f_3, y) \approx m_{y_i f_3}(y_i - E[y_i]). \tag{11}$$

For linear layers, max-pooling, and single-input activations, SHAP values admit closed-form solutions, making this a fast and principled approximation that eliminates the need to heuristically choose linearization strategies.

## 5 EXPERIMENTS AND RESULTS

The 10-feature configuration used: eye_open_l, eye_open_r, mouth_open, roll_def, yaw_rel, pitch_rel, lip_spread, brow_raise, gaze_ly, gaze_ry.

The 11-feature configuration (without yaw_rel) used: ear_l, ear_r, eye_open_l, eye_open_r, mouth_open, roll_def, pitch_rel, lip_spread, brow_raise, gaze_ly, gaze_ry.

The 12-feature configuration used: ear_l, ear_r, eye_open_l, eye_open_r, mouth_open, roll_def, yaw_rel, pitch_rel, lip_spread, brow_raise, gaze_ly, gaze_ry.

The 17-feature configuration used: ear_l, ear_r, eye_open_l, eye_open_r, mar, mouth_open, roll_def, yaw_rel, pitch_rel, lip_spread, brow_raise, sym_eye, sym_mouth, gaze_lx, gaze_ly, gaze_rx, gaze_ry.

During the SHAP-guided ablation experiments, we examined how the validation metrics change when individual features are removed from the input representation (Table 4). The 12-feature configuration produced the least suitable operating option: while recall increased, specificity dropped sharply, resulting in many false positives and a decrease in overall class-balanced performance and F1. This suggests that the features removed in the 12-feature setting helped suppress false alarms, and their absence made the classifier overly sensitive.

The 11-feature configuration uses the same feature set as the 12-feature experiment, but without yaw_rel, and it achieved the best imbalance-aware results among all tested variants. Compared to the 17-feature baseline, recall increased; compared to the 12-feature configuration, recall remained at the same high level; and compared to the 10-feature configuration, recall was higher, while balanced metrics (balanced accuracy and MCC) also improved. This pattern indicates that yaw_rel likely introduces substantial noise, and removing it leads to a better balance between the classes, while still requiring an operating-point choice that reflects whether sensitivity or false-alarm suppression is prioritized. The SHAP class-conditional importance plot for this best-performing 11-feature configuration is shown in Fig. 4.

In contrast, the 10-feature configuration retained yaw_rel but removed ear_l and ear_r. This setting produced a more conservative operating profile with higher specificity and fewer false positives, but lower recall and weaker imbalance-aware metrics (balanced accuracy and MCC) compared to the 11-feature configuration. Practically, the 11-feature configuration is preferable when sensitivity to reaction-present cases is prioritized under class imbalance, whereas the 10-feature configuration is preferable when limiting false positives is more important.

## 6 DISCUSSION

The Transformer architecture was chosen primarily due to its ability to model long temporal dependencies. During the design stage, the final duration of the videos was not strictly fixed, and we aimed to select an architecture capable of handling longer sequences if necessary. Even the current data (5 min x 25 FPS). Nevertheless, we agree that comparisons with alternative sequence models (e.g., LSTM, GRU, or temporal convolutional networks) might provide valuable insight, and such comparisons are planned for future experiments.

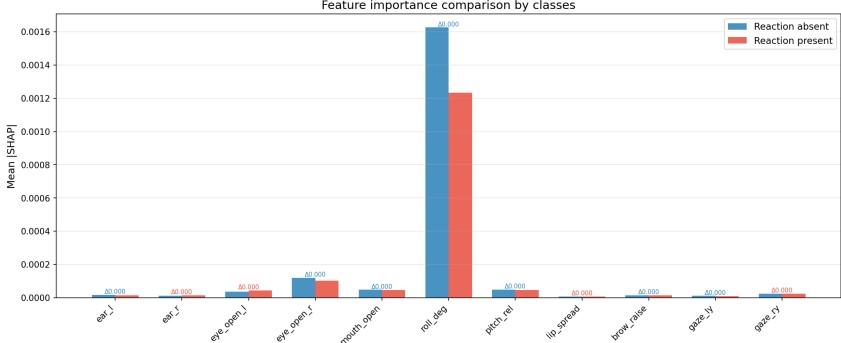

Figure 4: Class-conditional feature importance based on mean absolute SHAP values, comparing the average contribution of each feature to the model output for reaction-present and reaction-absent cases.

Table 4: Comparison of SHAP-guided feature pruning experiments (validation set $n = 44$, threshold policy: Youden).

| SET | BACC | PREC | REC | F1 | SPEC | MCC | PR | BRIER |
|---|---|---|---|---|---|---|---|---|
| 17 f (baseline) | 0.7048 | 0.4000 | 0.6667 | 0.5000 | 0.7429 | 0.3485 | 0.4330 | 0.1505 |
| **11 f (w/o yaw_rel)** | 0.7730 | 0.4000 | 0.8889 | 0.5517 | 0.6571 | 0.4423 | 0.4927 | 0.1529 |
| 10 f (w/o ear_l&r) | 0.7190 | 0.4286 | 0.6667 | 0.5217 | 0.7714 | 0.3794 | 0.4387 | 0.1484 |
| 12 f | 0.7016 | 0.3200 | 0.8889 | 0.4706 | 0.5143 | 0.3283 | 0.4349 | 0.1482 |

## 7 CONCLUSION

This work investigated SHAP-guided feature ablation for an interpretable facial time-series pipeline that maps FaceMesh dynamics to a compact set of semantically meaningful features and applies a Transformer-based classifier for binary reaction detection. Using Deep SHAP attributions, we assessed how removing individual features changes validation metrics and shifts the operating trade-off between missed reactions and false alarms.

The experiments show that reducing the feature set can preserve or improve prediction ability, but the outcome strongly depends on which features are removed. Excluding yaw_rel produced the strongest balanced results (higher balanced accuracy, MCC, and PR-AUC), suggesting that this feature introduces noise that harms class-balanced discrimination. For our imbalanced dataset, the 11-feature configuration (the 12-feature set without yaw_rel) provides the most appropriate overall operating point among all tested variants. In contrast, the 12-feature configuration resulted in an overly sensitive model with degraded specificity and a higher false-positive rate. Another competitive configuration (10 features) achieved a more conservative operating profile with higher specificity but lower recall, indicating that the preferred feature set depends on whether sensitivity or false-alarm suppression is prioritized. Overall, SHAP-based analysis served not only as a post-hoc explanation tool but also as a practical mechanism for feature set refinement and model selection.

Beyond SHAP-guided feature pruning in a manually constructed feature space, a natural next step is to address redundancy at the level of the original landmark representation. The 468 facial landmarks leads to more than 800 derived points on different axes, and consequently some features may be strongly correlated or nearly linearly dependent. Future work will therefore explore dimensionality reduction at the preprocessing stage using Principal Component Analysis (PCA). In particular, we plan to project the features into an orthonormal space, estimate the number of principal components required to preserve, for example, 95% of the original variance, and compare these components with the features identified as important by SHAP. The Transformer model will then be trained on the reduced PCA representation to evaluate potential gains in generalization, stability, and computational efficiency.

In addition, we plan to extend the current unimodal video-based setup toward a multimodal architecture. Separate models will be trained for video and audio streams, and their predictions will be integrated through a higher-level classifier, such as an MLP-based fusion module. This approach are expected to improve robustness and detection accuracy compared to single-modality configurations.

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
