# OpenReview forum: "SHAP-Guided Feature Selection and Model Simplification for Facial Time-Series Reaction Detection"
_mathai.club/MathAI/2026/Conference — 2026 Oral_

### Official Review · Reviewer_yJPd · 2026-03-11
**Review of "SHAP-Guided Feature Selection and Model Simplification for Facial Time-Series Reaction Detection."**

**Rating:** 4
**Confidence:** 5

**Review:**

This paper builds a complete pipeline for detecting facial reactions from video. The authors extract 17 hand-crafted features from FaceMesh landmarks, train a Transformer classifier, and use DeepSHAP to analyze which features matter most. They then prune low-importance features and retrain, comparing 17-feature, 12-feature, 11-feature, and 10-feature versions. The 11-feature set (removing "yawrel") performed best on balanced accuracy and MCC.

I actually liked the detailed and reproducible engineering pipeline. Along with that, unlike many XAI papers that stop at "here are the important features," the authors actually act on the SHAP values by iteratively pruning features and retraining, as shown in Table 4 with exact metrics, and the clear documentation makes it invaluable for anyone trying to build upon this work.

Despite my above words, I chose this rating because the paper fundamentally does not belong at MathAI 2026. The reasons are as follows:

First, the paper contains zero mathematical contribution. The conference mission is explicitly dedicated to "new mathematical methods and approaches" and "breakthrough mathematical solutions." This paper introduces no new theorems, no proofs, no lemmas, and no mathematical frameworks. The entirety of Section 4 is a textbook restatement of Lundberg & Lee 2017's existing work on SHAP, with Equations (1) through (11) reproduced directly from that paper. This is exposition, not contribution.

Second, while the CFP includes "game theory for explaining and interpreting" as a topic area, the paper makes no contribution to explanation methodology or to the mathematical understanding of game-theoretic explanation. The authors use DeepSHAP exactly as implemented in existing libraries with no adaptation, no methodological innovation, and no analysis of explanation quality (fidelity, stability, sensitivity). The paper does not ask or answer any research question about explanation itself—it simply uses SHAP as a tool to rank features for pruning. There is no comparison to alternative explanation methods (LIME, Integrated Gradients, Attention Rollout), no analysis of whether SHAP attributions are reliable for this specific Transformer architecture and facial time-series data, and no mathematical analysis of the properties of SHAP values in this context. The contribution is in the engineering loop (train → prune → retrain), not in advancing the mathematical foundations of game-theoretic explanation.

Third, the core "SHAP-guided feature selection procedure" is purely heuristic. It is a simple loop: train → compute mean absolute SHAP → remove bottom features → retrain. There is no theoretical justification for why this procedure should converge to an optimal feature set, no proof that it minimizes any objective function, no comparison to alternative methods like LASSO or forward selection, and no analysis of feature stability across different train/test splits. This is engineering intuition, not mathematical innovation.

Fourth, the results are entirely dataset-specific and not generalizable. The finding that "removing yawrel improved performance" applies only to this specific dataset (47 subjects, filmed under particular conditions), these specific 17 hand-crafted features, this specific Transformer architecture (d_model=128, 2 layers), and this specific binary classification task. The paper offers no theoretical reason why yawrel is noisy, no mathematical analysis of its properties, and no guarantee that this insight would transfer to any other dataset or task.

Finally, there is a fundamental misalignment with conference scope. MathAI 2026's mission is to create a platform for "leading experts in the field of mathematics" to form a "formal basis" for AI systems. This paper is written by and for engineers, contributes to applied ML practice rather than mathematical foundations, and would be perfectly appropriate at computer vision venues such as FG, ACII, ICMI, or CVPR workshops. It has no place at a conference dedicated to mathematical advancements in AI.

In conclusion, this is a well-executed piece of applied engineering with excellent reproducibility and practical value. However, it contains zero mathematical novelty and fundamentally does not belong at MathAI 2026. It is a clear scope rejection.

---

### Official Review · Reviewer_DZqG · 2026-03-11
**An interpretable facial reaction detection pipeline using semantic features and SHAP-guided feature pruning, though the contribution is incremental and requires stronger experimental validation.**

**Rating:** 6
**Confidence:** 5

**Review:**

Summary

The paper proposes an interpretable pipeline for facial time-series reaction detection using FaceMesh landmarks. Instead of using raw high-dimensional landmark trajectories, the authors design 17 semantic facial dynamics features (eye openness, mouth dynamics, symmetry, motion descriptors) and train a Transformer encoder classifier to detect reactions. To improve interpretability and efficiency, the study introduces an iterative SHAP-guided feature selection framework that prunes low-impact features while preserving predictive performance.

Strengths

1. Interpretability-focused design
The use of semantically meaningful facial features rather than raw landmarks improves interpretability and aligns well with psychological facial analysis frameworks.

2. Practical dimensionality reduction approach
Reducing FaceMesh inputs (468 landmarks) to a compact representation is sensible and can improve robustness and computational efficiency.

3. Novel use of SHAP for iterative feature pruning
Using SHAP not just for explanation but as a model simplification loop (train → compute SHAP → prune → retrain) is a practical and reproducible idea.

4. Clear motivation
The paper clearly explains why raw landmark trajectories are problematic (high correlation, high dimensionality, nuisance factors).

5. Good methodological clarity
The pipeline and modeling choices (Transformer encoder for temporal signals) are appropriate and well motivated.

Weaknesses

1. Limited methodological novelty
While the SHAP-guided pruning loop is useful, it is conceptually incremental. Feature selection via SHAP importance is already common in ML interpretability literature.

2. Dataset description appears incomplete
The dataset section is very brief and does not clearly describe:

number of participants

recording conditions

dataset size

train/test split

labeling protocol

These are critical for reproducibility.

3. Lack of strong experimental evaluation
The paper should include:

comparison with raw landmark models

comparison with standard feature selection methods

ablation studies

statistical significance

4. Transformer justification could be stronger
It is not clear whether the sequence length truly requires attention mechanisms vs simpler models (LSTM, GRU, temporal CNN).

5. Reaction definition ambiguity
The target variable “reaction vs no reaction” needs clearer operational definition and labeling process.

---

### Official Review · Reviewer_qpiy · 2026-03-13
**An interpretable approach to facial reaction detection with moderate empirical support**

**Rating:** 6
**Confidence:** 3

**Review:**

1) Summary

The paper studies binary facial reaction detection in constrained question–answer episodes. The authors argue that raw FaceMesh landmark trajectories are high-dimensional, correlated, and difficult to interpret, so they map them into 17 semantically meaningful facial-dynamics features and train a Transformer-based classifier. The main claimed contribution is to use Deep SHAP not only for post-hoc explanation, but also for iterative feature selection and model simplification by pruning low-impact features and retraining the model.

The experimental section evaluates several reduced feature sets under class imbalance. The paper reports that one of the reduced 11-feature configurations achieves the best imbalance-aware validation performance among the tested variants, outperforming the 17-feature baseline in balanced accuracy and MCC.

2) Strengths

The paper addresses a practically relevant problem and presents the full pipeline clearly, from annotation conversion and clip extraction to feature construction, model training, and evaluation. The manuscript is generally easy to follow and the overall structure is coherent.

A notable positive aspect is the emphasis on interpretability. Instead of directly training on dense landmark coordinates, the paper constructs a compact and semantically meaningful feature space, and then uses SHAP to analyze feature contributions. The evaluation also includes several metrics that are appropriate for imbalanced binary classification, rather than relying on a single headline number.

3) Weaknesses

The main concern is the level of scientific novelty. In its current form, the work appears to combine existing ingredients — hand-crafted facial features, a lightweight Transformer encoder, and Deep SHAP-based analysis — into an application pipeline. While this may be practically useful, the paper does not yet make a sufficiently strong case that the proposed method itself is novel beyond applying known tools in this domain.

Another issue is that some core claims are not fully matched by the presented experiments. The paper describes an iterative SHAP-guided staged reduction procedure, including a possible reduction down to 7 features, and motivates the approach partly by expected computational savings. However, the reported results only cover 17-, 12-, 11-, and 10-feature settings, and no explicit runtime or inference-cost measurements are presented.

4) Verdict

The paper presents a clear and practically motivated pipeline for interpretable facial reaction detection, and the use of SHAP for feature-level analysis is potentially useful in this setting. While the methodological novelty is limited and the empirical validation remains relatively narrow, the submission is reasonably structured and provides enough evidence to suggest practical value for the target venue. I therefore view it as marginally above the acceptance threshold, while also believing that stronger baselines, clearer validation protocol, and tighter support for the feature-selection claims would substantially strengthen the work.

---

### Decision · Program_Chairs · 2026-03-14

**Decision:**

Accept (Oral)

**Comment:**

Dear Author(s),

On behalf of the Program Committee of the International Conference on Mathematics of Artificial Intelligence (MathAI 2026), we are pleased to inform you that your paper has been accepted for an oral presentation at MathAI 2026.

Your paper was evaluated through a rigorous two-stage review process involving both automated screening and expert review by members of the Program Committee. The reviewers recognized the quality and contribution of your work.

Presentation details:

- Format: Oral presentation (15–20 minutes + 5 minutes Q&A)
- Mode: You may present either in person (offline) at the conference venue in Sirius, Russia, or remotely via Zoom. Please indicate your preferred mode when confirming your participation.
- Conference dates: Marh 30 - April 3, 2026
- Website: https://mathai.club

Next steps:

1. Please confirm your participation and presentation mode by replying to this email mathai.club@yandex.ru no later than March 15, 2026 18:00 Moscow time.
2. If you plan to attend in person, the organizing committee will provide accommodation details separately.
3. Please prepare your final camera-ready manuscript according to the formatting guidelines available at https://mathai.club and upload it to OpenReview by March 15, 2026 18:00 Moscow time.

Should you have any questions regarding the program, logistics, or your presentation slot, please do not hesitate to contact us.

We look forward to your contribution to MathAI 2026.

With kind regards,

MathAI 2026 Program Committee
International Conference on Mathematics of Artificial Intelligence
https://mathai.club
OpenReview: https://openreview.net/group?id=mathai.club/MathAI/2026/Conference
Telegram: https://t.me/MathAI_club
Email: mathai.club@yandex.ru